# Permafrost Nitrous Oxide Emissions Observed on a Landscape Scale Using Airborne Eddy Covariance Method

Jordan Wilkerson[1], Ronald Dobosy[2,3], David S. Sayres[4], Claire Healy[5], Edward Dumas[2,3], Bruce Baker[2], and James G. Anderson[1,4,5]

[1]Department of Chemistry and Chemical Biology, Harvard University, Cambridge, MA 02138, USA; [2]Atmospheric Turbulence and Diffusion Division, NOAA/ARL, Oak Ridge, TN 37830, USA; [3]Oak Ridge Associated Universities (ORAU), Oak Ridge, TN 37830, USA; [4]Paulson School of Engineering and Applied Sciences, Harvard University, Cambridge, MA 02138, USA; [5]Department of Earth and Planetary Sciences, Harvard University, 12 Oxford Street, Cambridge, MA 02138, USA.

*Correspondence to*: Jordan Wilkerson (jwilkerson@g.harvard.edu)

**Abstract.** The microbial by-product nitrous oxide ($N_2O$), a potent greenhouse gas and ozone depleting substance, has conventionally been assumed to have minimal emissions in permafrost regions. This assumption has been questioned by recent *in situ* studies demonstrating that, in fact, some geologic features in permafrost may have elevated emissions comparable to those of tropical soils. These recent studies, however, along with every known *in situ* study focused on permafrost $N_2O$ fluxes, have used chambers to examine small areas ($< 50$ $m^2$). In late August 2013, we used the airborne eddy covariance technique to make *in situ* $N_2O$ flux measurements over the North Slope of Alaska from a low-flying aircraft spanning a much larger area: around 310 $km^2$. We observed large variability of $N_2O$ fluxes with many areas exhibiting negligible emissions. Still, the daily mean averaged over our flight campaign was 3.8 (2.2-4.7) mg $N_2O$ $m^{-2}$ $d^{-1}$ with 90% confidence interval in parentheses. If these measurements are representative of the whole month, then the permafrost areas we observed emitted a total of around 0.04-0.09 g $m^{-2}$ for August, comparable to what is typically assumed to be the upper limit of yearly emissions for these regions.

## 1 Introduction

$N_2O$ is the third most influential anthropogenic greenhouse gas behind $CO_2$ and $CH_4$. Inert in the lowest atmospheric layer, $N_2O$ eventually rises into the stratosphere. There, photolysis and electronically excited oxygen atoms ($O(^1D)$) convert $N_2O$ to nitrogen oxides that catalytically deplete ozone. $N_2O$ is currently the dominant ozone-depleting substance anthropogenically emitted. It is expected to remain so throughout the entire 21st century (Ravishankara et al. 2009). Due to increased industrial processes and agricultural practices that rely on heavy fertilization, $N_2O$ concentrations have been steadily rising in the atmosphere (Park et al. 2012). With a Global Temperature-change Potential over a 100-year time scale ($GTP_{100}$) of 296, the climate system is more sensitive to changes in $N_2O$ concentrations than either of its carbon-based GHG counterparts (IPCC 2013).

While the global $N_2O$ budget can be divided into natural and anthropogenic sources, the two sectors have one thing in common: the primary mechanism of emission is denitrification by soil microbes (Syakila et al. 2011). For the anthropogenic sector, this primarily comes in the form of enhanced microbial activity in agricultural soils due to an imbalance between N fertilizer supply and crop uptake (Syakila et al. 2011). For the natural sector, tropical soils are considered the largest source of $N_2O$ (Zhuang et al. 2012). Meanwhile, $N_2O$ emissions from permafrost-laden regions have long been assumed to be negligible (Martikainen et al. 1993; Potter et al. 1996) and are ignored in current $N_2O$ budgets (Anderson et al. 2010; Zhuang et al. 2012). This is largely because higher latitudes are considered nitrogen-limited and biogeochemically inactive relative to the tropics (Zhuang et al. 2012).

As a result, permafrost $N_2O$ emissions have not received anywhere near the same level of monitoring as either $CO_2$ or $CH_4$. In Arctic tundra regions, an array of flux towers provides continuous measurements for $CO_2$ flux and increasingly for $CH_4$ flux as well, while no measurement like this exists for $N_2O$ (McGuire et al. 2012). This is reasonable if permafrost $N_2O$ emissions are truly negligible as is assumed. However, recent *in situ* measurements of permafrost soils in Russian tundra and northern Finland (Repo et al. 2009; Marushchak et al. 2011) have found several geologic formations that may emit $N_2O$ fluxes comparable to tropical soil emissions (Zhuang et al. 2012). These formations include bare peat surfaces and thaw-induced permafrost collapse known as thermokarst. Elevated production of $N_2O$ in soil has also been observed in thermokarst features on the Alaskan North

Slope (Abbott et al. 2015). All of these studies reported that these trends were sustained throughout the growing period. Furthermore, mesocosm studies in Finnish Lapland along with separate laboratory studies suggest that thawing permafrost further increases $N_2O$ production (Elberling et al. 2010; Voigt et al. 2017). Permafrost contains ~73 billion tons N in the upper 3 m of its soils (Harden et al. 2012). Considering this, better understanding the magnitude of current $N_2O$ emissions from Arctic surfaces is crucial given that the current rate of thaw is expected to continue or increase over the next century (Jones et al. 2016; Borge et al. 2017).

The past studies on permafrost $N_2O$ emissions have provided insight into the mechanisms of the gas's production and subsequent release into the atmosphere. The studies have been either laboratory studies or ground-based chamber studies. In general, chamber studies have the advantage of observing the same site for relatively long time periods. Additional variables (e.g. pH, water saturation) can be monitored as well, which are crucial to understanding how that environment might influence the observed extent of $N_2O$ emissions. However, each chamber covers around 1 $m^2$, and a feasible chamber study can only entail a limited number of sites. Consequently, past observations have covered extremely small areas – less than 50 $m^2$ (Repo et al. 2009; Marushchak et al. 2011; Yang et al. 2018).

The Arctic tundra entails a mosaic of different land surfaces. This high spatial variability makes it challenging to use chamber flux observations to draw conclusions about the landscape and regional scale (McGuire et al. 2012). This is especially true for $N_2O$ from permafrost because 1) there are not rigorously defined emission factors for permafrost land surfaces suspected to be significant emitters of $N_2O$, and 2) soil $N_2O$ emissions are known to exhibit particularly high spatial variability (Butterbach-Bahl et al. 2013). Therefore, it is uncertain whether the recent research findings showing significant permafrost $N_2O$ production and emission (chambers, soil analyses, and laboratory studies) are reflective of a larger trend in the Arctic tundra or if these are just isolated incidents.

An alternative approach that can help answer this question is airborne eddy covariance (EC) measurement. The nature of an airborne study is to provide a spatial survey of the prevalence and spatial distribution of such high-emission locations along with any other distributed sources in an area difficult of access. An advantage over chamber measurements is that an airborne campaign folds in the mosaic of land surfaces present, so fluxes can be directly integrated over areas with high spatial variability (McGuire et al. 2012). Landscapes deemed vulnerable to thaw-induced $N_2O$ emissions, based on Arctic mesocosm studies, cover about one fourth of the Arctic/sub-Arctic (Voigt et al. 2017). One of those vulnerable areas is the Alaskan North Slope, which is the focus of this study. To get a landscape-scale estimate of the magnitude of permafrost $N_2O$ emissions during late summer, we measured $N_2O$ fluxes over the North Slope in late August 2013 using the airborne EC technique.

Recent developments of fast-response $N_2O$ gas analyzers has made feasible the application of the EC technique to nitrous oxide (Rannik et al. 2015). EC flux towers have been used to measure $N_2O$ in several regions other than Arctic tundra. The first $N_2O$ EC flux tower measurements were published over a decade ago, using quantum cascade laser (QCL) spectroscopy (Kroon et al. 2007; Eugster et al. 2007). The specific QCL spectroscopic method used in this study to measure $N_2O$ mixing ratios, known as off-axis integrated cavity output spectroscopy (OA-ICOS), has also been applied to $N_2O$ EC flux measurements before (Zona et al. 2013). Furthermore, comparative measurements of $N_2O$ fluxes from chambers and EC towers have been performed in a drained peatland forest, and the two techniques showed reasonably good agreement (Pihlatie et al. 2010). While the airborne application of the EC technique has not previously been used with nitrous oxide, airborne EC has been used to measure fluxes of other trace gases at least over the past 30 years (Sellers et al. 1997). From multiple comparison studies, the airborne version of EC is considered as reliable as EC from a flux tower, the difference being that it averages over space instead of time (Mahrt et al. 1998; Gioli et al. 2004). The North Slope's large flat terrain makes it particularly suitable for airborne EC measurements (Hensen et al. 2013; Sayres et al. 2017).

**2 Methods**

To evaluate landscape nitrous oxide fluxes in the North Slope, the Flux Observations of Carbon from an Airborne Laboratory (FOCAL) system (shown in Fig. 1) was flown out of Deadhorse Airport, Prudhoe Bay, AK. Though its name came from its ability to measure $CO_2$ and $CH_4$ fluxes, it can simultaneously measure $N_2O$ flux. Measurements were made over five separate flights in several regions of the North Slope from 2013 August 25-28. The measurements entailed a cumulative path length of 884 km and approximate area coverage of 310 $km^2$ (Fig. 2, Table 1). Flights consisted of either repeated flight tracks near a $CH_4/H_2O$ EC flux tower or 50 x 50 km grid patterns. In each flight, the flux calculations were restricted to straight segments flown below 50 m AGL. For the present study, segment sections over the open ocean were also excised.

The low-flying aircraft flown in the campaign, a Diamond DA-42 from Aurora Flight Sciences, housed the two main components for flux measurements (Fig. 1): a turbulence probe and a custom-built IR spectrometer measuring water vapor, $CH_4$, and $N_2O$ at a rate of 10 times per second (10 Hz). These two components were used to measure the EC fluxes of $N_2O$, $CH_4$, and $H_2O$ during the 2013 campaign.

The flights near the flux tower were performed to compare the airborne $CH_4$ and $H_2O$ flux measurements with those from the EC flux tower (Dobosy et al. 2017). The $CH_4$ and $H_2O$ fluxes agreed with the ground measurement, and the $CH_4$ fluxes are consistent with other observed summertime permafrost $CH_4$ emissions reported in the scientific literature (see Sayres et al. 2017). The only difference in the airborne flux measurements between $CH_4$, $H_2O$, and $N_2O$ is the particular absorption feature used within the observed spectral region of the IR instrument, as further discussed in Section 2.2 (Fig. 3).

## 2.1 BAT Probe Description and Calibration

The three wind components were measured using the Best Airborne Turbulence (BAT) probe developed by the National Oceanic and Atmospheric Administration/Atmospheric Turbulence and Diffusion Division (NOAA/ATDD) in collaboration with Airborne Research Australia (Crawford et al. 1993; Dobosy et al. 2013). The BAT probe also recorded ambient temperature and pressure measurements, which were used to determine dry air density. The aircraft was equipped with a radar altimeter, which in conjunction with three-component wind velocity measurements, was used for footprint calculations. These calculations were performed over 60 m segments along the flight track (Sayres et al. 2017). The footprints, representing the area from which the observed fluxes originated, were used to estimate the total area measured and identify which land classes were measured (Table 1).

The BAT probe, developed in the 1990s, is a type of gust probe consisting of a hemispherical head, 15.5 cm in diameter, with ports at selected positions on the hemisphere to sample the pressure distribution. A gust probe functions similarly to a typical pitot-static system but includes additional pressure measurements to sense the direction of the incoming flow along with its speed. The direction is specified in two perpendicular components called angles of attack and sideslip, which rarely exceed $\pm10°$ in balanced flight (Leise et al. 2013). The BAT probe differs from other gust probes in having a larger head to accommodate accelerometers and pressure sensors directly in the head simplifying the physical and mathematical system needed to determine turbulent wind. It also has nine ports instead of the usual five found in traditional gust-probe systems. These additional four ports measure the ambient atmospheric pressure apart from small adjustments for nonzero attack and sideslip angles. Wind is sampled at 1000 Hz, filtered to control aliasing, and subsampled at 50 Hz.

The BAT probe configured for the FOCAL campaign (with the gas inlets in place) was characterized in a wind tunnel (Dobosy et al. 2013) following on an earlier wind-tunnel test of a similar unit in Indiana, USA (Garman et al. 2006). Its overall precision for wind is $\pm0.1$ m s$^{-1}$. With the entire instrument system assembled, standard-practice calibration maneuvers were flown in smooth air to establish the values of the tuning parameters for temperature, pressure, and wind measurement (Vellinga et al. 2013). Following the usual practice, we also made a calibration flight in smooth air on August 27 toward the end of the campaign (Sayres et al. 2017). Plots and comparison of spectra, cospectra, and time series for each flight provide tests of the quality of the data and of the processing through all intermediate steps.

## 2.2 $N_2O$ Instrument Description and Calibration

The gas inlet for the $N_2O$ instrument is located on the BAT probe housing, 8 cm aft of the probe's hemispherical face, where ambient pressure and temperature measurements are made. The custom-built IR instrument uses Off-Axis Integrated Cavity Output Spectroscopy (OA-ICOS) to simultaneously measure $H_2O$, $CH_4$, and $N_2O$ (Fig. 4 & Fig. S1). The light source is a distributed feedback (DFB) continuous-wave quantum cascade laser (QCL) (Hamamatsu, LC0349). The laser tunes from 1292.5 to 1293.3 cm$^{-1}$ in 1.6 milliseconds. This region contains absorption features for $H_2O$, $CH_4$, and $N_2O$ (Fig. 3). Before the light enters the optical cavity, a beam-splitter diverts some of it through a Ge etalon. The etalon measures the rate at which the laser is tuning across the wavelength region, which is used to determine the width of the absorption lines. These components are all housed in the laser pressure vessel (Healy 2016).

The detection cell is a 25-cm length optical cavity composed of two high-reflectivity ZnSe mirrors (LohnStar Optics, R = 0.9996), which creates an effective path length of ~625 m. After leaving the cavity, the light enters the detector pressure vessel where it is focused onto a Stirling-cooled HgCdTe photoconductive detector (InfraRed Associates, Inc., MCT-12-2.05C). The detector system samples the light at 100 MHz and averages the readings to produce raw spectra with 1900 samples each. These spectra are then co-added to produce 1 spectrum every 0.1 sec and are stored on the flight computer.

Sample flow through the optical cavity is maintained in flight with a dry scroll pump that flushes the cell 17 times per second. The optical cavity is temperature- and pressure-controlled to T = $303.70 \pm 0.05$ K and p = $59.26 \pm 0.01$ Torr to allow conversion from concentration (moles cm$^{-3}$) to mixing ratio. The cell temperature is measured by averaging the output of two 1 M$\Omega$ thermistors (General Electric, Type B) located within the cell. These were calibrated against a platinum primary standard. The cell is heated by polyimide thermofoil heaters, which are located along the cell exterior. The cell pressure is measured with a dual-headed absolute pressure transducer (MKS, D27D) and is controlled by a proportional solenoid valve. The valve is coupled with a pressure control board that uses the pressure transducer as feedback on the valve orifice's position (Fig. 1a) (Healy 2016).

Measurement of $H_2O$ was calibrated using a dry-air tank coupled with a bubbler flow system as described in Weinstock et al. 2009. The $H_2O$ measurements were used to account for dilution and water-broadening effects on the $N_2O$ absorption feature and to convert the mixing ratio from moles per mole of total air to moles per mole of dry air for flux computation (Webb et al. 1980, Gu et al. 2012). The broadening coefficients were determined using the approach described in Rella 2010.

Periodic in-flight calibrations were performed to track and correct for drift over the course of the flight (2 calibration cycles per flight). These were performed using a secondary standard (277 ppbv $N_2O$) calibrated in lab to a WMO standard (Sayres et al. 2017). Before and after the campaign, calibrations were also conducted in lab using two primary WMO standards and a synthetic air tank (containing no $N_2O$) to calibrate the absorption coefficient and check for linearity. The short-term precision of the ICOS

instrument for $N_2O$ mixing ratios is determined using

$$\sigma = \sigma_{1s} f_s^{-1/2} \tag{1}$$

where $\sigma_{1s}$ is the 1-second standard deviation for in-flight calibration data collected during that particular flight, and $f_s$ is the

sampling frequency in Hz (Kroon et al. 2007). Optical alignment was occasionally adjusted between flight days resulting in an $N_2O$ precision range over the five flights of $\sigma = 0.27\text{-}0.58$ ppb $Hz^{-1/2}$ (Table S1). This is close to the recommended precision for $N_2O$ EC flux measurements as determined by previous studies evaluating the application of the EC technique to this particular trace gas; these groups also used QCL spectroscopy to measure $N_2O$ mixing ratios (Kroon et al. 2007; Eugster et al. 2007).

**2.3 Airborne EC Flux Calculations**

The Airborne EC method relies on the fact that gases like $H_2O$, $CH_4$, and $N_2O$ emitted from the surface are transported upward into the atmospheric boundary layer by turbulent eddies. On average, upward flux occurs when updrafts are, more often than not, enriched in the transported gas relative to downdrafts. The covariance of vertical wind velocity with gas concentration is thus positive for upward flux, negative for downward.

To determine the covariance between vertical wind velocity $w$ and $N_2O$ mixing ratios $c$, we first separate each variable into changes associated with large-scale air motion (i.e. advection) and small-scale air motion (i.e. turbulence) (e.g. $w = \bar{w} + w'$). We separated the two scales by fitting fourth-order polynomials to the measurements of $w$ and $c$ made along each individual straight leg of each flight (Fig. 2). The fit itself incorporates the larger scale trends (e.g. $\bar{w}$), which are subtracted from the data. The remaining residuals from this fit are the turbulent quantities of interest (e.g. $w'$) (Foken 2008).


Multiplying $w$ by the density of dry air $\rho_d$ and extracting the residual as discussed above, one obtains the turbulent dry air mass flux. The covariance of this dry-air flux $(\rho_d w)'$ with the turbulent mixing ratio $c'$ then yields the trace-gas flux of interest by the general EC approach (Webb et al. 1980, Gu et al. 2012):

$$F = \overline{(\rho_d w)' c'} \tag{2}$$

As previously mentioned, airborne EC measurements average over space instead of time. Accordingly, we compute the $N_2O$ fluxes (along with $CH_4$ and $H_2O$ fluxes) using the general equation for airborne EC flux calculations:

$$F = \frac{\sum_{k=1}^{N} (\rho_d w)'_k c'_k V_k}{\sum_{k=1}^{N} V_k} \tag{3}$$


where $V$ is the airspeed of the aircraft, and the other variables are defined as in Eq. (2) (Sayres et al. 2017; Dobosy et al. 2017). The true airspeed $V = dl/dt$ is included in Eq. (3) to convert the variable of integration from time to space because the raw data are recorded at uniformly spaced time intervals (every 0.1 s) (Crawford et al. 1993). A number $N$ of samples is being averaged, the denominator yielding their cumulative path length through the air.


Air density and vertical wind velocity $w$ from the BAT probe are filtered and then subsampled at 10 Hz to match the measurement frequency for $c_{N2O}$, $c_{CH4}$, and $c_{H2O}$. Because the BAT probe observes a specific packet of air before the spectrometer does, a correction for the lag is applied to the data. The lag time from the gas inlet to the optical cavity was measured in the laboratory to be around 0.55 s. The lag between the BAT-probe measurements and those of the ICOS instrument

in flight were determined by cross-correlation analysis of $w$ and $c_{CH4}$. They varied between 0.4 and 1.2 s. Methane was used as a proxy for $N_2O$ to determine the lag because of its stronger signal.

Computation of dry air density uses the measured dry-air mixing ratio of $H_2O$. Turbulent quantities required for the footprint model are then computed by summing Eq. (3) over each flight segment. The mean $N_2O$ fluxes displayed in Table 2 are computed

by summing Eq. (3) over the multiple segments of each flight depicted in Fig. 2, excluding flight-path sections over coastal waters.

**2.4 Flux Uncertainty Analysis**

All confidence intervals reported in Table 2 and Table S1 are derived using bootstrap resampling (Dobosy et al. 2017), not from $w$ and $c_{N2O}$ individually but from flux fragments (Sayres et al. 2017; Dobosy et al. 2017). These fragments are short (typically 1 s) blocks of integrated data that include the integrals of the three wind components, the height above ground, and the cross products of turbulent departure quantities from Eq. (3). All are integrated as above over the path through the air rather than time. They are each about 60 m long varying slightly due to small airspeed changes. These measurements, and therefore the corresponding confidence intervals, contain both environmental variability and variability arising from instrumental noise.

The fragments are serially correlated, and their means, trends, and variances are heterogeneous on scales greater than the 6 km found by ogive analysis to belong to turbulence. A procedure described by Mudelsee decomposes such partially determined, autocorrelated, and variably spaced data streams using the equation,

$$X(S) = T(S) + \sigma(S)R(S). \tag{4}$$

Here $X(S)$ is a random-variable function over the path length S defined at irregular intervals (Mudelsee et al. 2010). The $T(S)$ and $\sigma(S)$ are the deterministic trend and variance of $X(S)$ for each $S$, and $R(S)$ is a serially correlated random-variable series having zero mean and unit variance. The $T(S)$ and $\sigma(S)$ for the current analysis are estimated as overlapping averages and variances of the measured fragments taken over 6 km, as determined by ogive analysis. They are evaluated at intervals of 1 km along the track and treated as fixed throughout the rest of the process. These larger scales can be treated as determinable from some (mesoscale) model.

The serial correlation of the random series $R(S)$ is removed by a first-order Markov model, the inverse of a first-order causal filter (Dobosy et al. 2017). The resulting decorrelated series is (ideally) independent and "weakly" homogeneous (i.e., it has zero mean and unit variance). As such, it is suitable for bootstrap resampling. A resample size of 80,000 random decorrelated sequences, each the same length as the original set of fragments was drawn. The explained portion of the variance was then reapplied to each new resample using a process that is the reverse of the process of its removal, thus providing an ensemble of 80,000 new potential outcomes of the original experiment. The confidence intervals for $N_2O$ flux were determined from the distribution of this population of reconstituted potential outcomes.

A Student's $t$-test was also used to evaluate whether the Pearson's correlation coefficient for $w'$ and $c_{N2O}'$, hence the $N_2O$ flux, differs significantly from a random (zero) correlation (Eugster et al. 2015). Because the atmospheric data stream is serially correlated, as noted above, the $N$ samples do not represent the total number of *independent* samples $n$. The number of independent samples is determined by:

$$n \cong N \frac{1-\rho_1}{1+\rho_1} \tag{5}$$

where $\rho_1$ is the lag-1 autocorrelation coefficient (Eugster et al. 2015). The conclusions of this test are folded into Table 2 in the Results and Discussion section.

**2.5 Footprint and Land Class Determination**

Following Sayres et al. 2017, the model developed by Kljun et al. 2004 is used to derive a footprint for each 60-m flux fragment. The wind and height above ground level are averaged over the fragment, and the required turbulence quantities are averaged over the flight leg containing the fragment. This procedure accounts for variations in mean wind and height above ground along the track while using the longer average required for flux computation. The footprints are used to estimate each flux fragment's related source area on the surface. The set union of these source areas constitutes the total area covered by each flight. Each flux fragment's coverage area is estimated by multiplying the length accounting for 90% of the crosswind-integrated probability in the footprint by the path length of the respective flux fragment. For Flights 25.18 and 27.19, the aircraft sometimes flies over the same path multiple times during the same flight. In these cases, the observed area is only counted once. These areas are summed for all of the fragments in each flight (Table 2).

For determining the land classes associated with the flux footprints, we use a land cover map developed by the North Slope Science Initiative (NSSI). There were 24 land classes used for the NSSI classification scheme (NSSI 2013). However, for our land classification, the following land classes are conflated: Tussock Tundra and Tussock Shrub Tundra; Freshwater Marsh Arctophila Fulva and Freshwater Marsh Carex Aquatillis; Dwarf Shrub – Dryas and Dwarf Shrub – Other; Ice/Snow and Open Water.

# 3 Results

## 3.1 Cospectra and Ogives

A cospectrum, the spectral decomposition of the covariance of the vertical wind velocity and trace gas mixing ratio, reveals the contribution to the overall flux from turbulent eddies as a function of their size. Starting from the smallest scales, the cospectrum
normally increases to a maximum and then returns to near zero, sometimes increasing again at still larger scales. The cumulative integral of the cospectrum up to its first return to zero is known as the ogive. The cospectra and ogives averaged over the entire campaign are shown for $H_2O$, $CH_4$, and $N_2O$ in Fig. 5.

The usual cospectrum has the shape of the $H_2O$ flux (blue). This cospectrum reaches a peak around 300 m above which the
incremental contribution to the flux declines with increasing eddy scale reaching near zero in this case at about 6 km. This length is taken to be the largest scale of boundary-layer turbulence for $H_2O$. It is, therefore, the minimum averaging length for $H_2O$ fluxes. For the ogive, this point corresponds to the inflection point (zero slope). The cospectrum and ogive curves are normalized by the value of the ogive at its inflection point. The ogive thus reaches unity at its inflection point, where it is proportional to the mean flux density ($g\ m^{-2}\ s^{-1}$) of $H_2O$.

The cospectra of the remaining two gases ($CH_4$ & $N_2O$) follow the same pattern but with considerably more scatter due to the weaker flux of these gases, among other things. The ogive of $N_2O$ in particular shows notably strong contributions from the smaller scales (below 80 m) and the larger scales (above 500 m) perhaps resulting from a spotty distribution of sources (e.g., Fig. 6). The maximum turbulent scales for $CH_4$ and $N_2O$ were taken from these ogives to be 4 km and 6 km, respectively.

## 3.2 Spatial Distribution of $N_2O$ Flux

Fig. 6 shows a spatial map of $N_2O$ emissions measured during Flight 28.10. The individual points represent running averages obtained by summing Eq. (3) over 6 km paths with 3 km overlap. The choice of 6-km averaging length for Flight 28.10 was
determined by ogive analysis as discussed above (Foken 2008).

The detection limit for observing these 6-km average $N_2O$ fluxes was estimated by computing the running averages using Eq. (3) as described above but replacing the measured environmental $c_{N2O}$ with a synthetic vector of the same length (~55 min.). The synthetic vector was assembled by random resampling with replacement from 3 minutes (1800 samples) of $N_2O$ mixing ratio
obtained during the same flight from a cylinder having known $N_2O$ mixing ratio. All other data streams, such as $\rho_d$ and $w$, containing both instrumental and environmental variability, remained unchanged. These 6-km running averages composed of calibration data had mean $0.0 \pm 0.05\ \mu g\ N_2O\ m^{-2}\ s^{-1}$ (standard deviation). We use $2\sigma$ to define the instrumental uncertainty of the 6-km averages and treat 6-km average values between $\pm 0.1\ \mu g\ N_2O\ m^{-2}\ s^{-1}$ in Fig. 6 as indistinguishable from zero.

While Flight 28.10 had a significant overall average emission (Table 2), Fig. 6 illustrates that much of that emission arises in multiple small-scale domains (commonly referred to as 'hot spots') with approximately half of the 6-km means being indistinguishable from zero given the instrumental uncertainty described above. This hot-spot pattern is true for the other flights as well. This highlights the spatiotemporal variability characteristic of soil $N_2O$ emissions and could help explain why permafrost $N_2O$ emissions studies, which are sparse and rely on extremely small spatial sampling when done, often detect no significant flux
(Kroon et al. 2007; Butterbach-Bahl et al. 2013).

## 3.3 $N_2O$ Flux Averages

The whole-flight spatially averaged $N_2O$ fluxes for each flight and the approximate surface area covered are shown in Table 2. Several of the averages are higher than have been assumed for $N_2O$ emissions at these latitudes. For example, the average $N_2O$ flux from tundra is considered to be around $0.005\ \mu g\ N_2O\ m^{-2}\ s^{-1}$ (Potter et al. 1996). By contrast, the whole-flight average from Flight 28.10, dominantly from tussock tundra (Table 1), was $0.104\ \mu g\ N_2O\ m^{-2}\ s^{-1}$, 20 times higher than the assumed value. Of the five flights, there are two flights where the average agrees with the expectation of negligible emissions (Flights 27.11 and
27.19). The flight where we observed the lowest average $N_2O$ flux, Flight 27.11, covered land surfaces significantly more waterlogged than the other flights (Table 1). This observation is consistent with the established understanding that water saturation acts as a suppressant of nitrous oxide emissions because $N_2O$ is instead anaerobically processed into $N_2$ by nitrous oxide reductase, an enzyme hypothesized to be inhibited by $O_2$ (Morley et al. 2008; Butterbach-Bahl et al. 2013.

Explaining the low mean flux for Flight 27.19 is less straightforward. Even though the path of Flight 27.19 was in the same proximity as Flights 25.18 and 28.15, Flight 27.19 has a noticeably lower average than the other two, which have similar mean fluxes (Table 2). Because this was an evening flight under reduced solar radiation, the boundary layer may have been too shallow to communicate the full emission of $N_2O$ from the surface to the flight level where it could be measured (Sayres et al. 2017). Separately, Table 1 shows that the ordering of the land classes measured was the same for all three flights. However, Flight

27.19's contribution from lakes and freshwater marsh was higher than from the other flights in that area (Sayres et al. 2017). Therefore, the comparatively lower average could also be a consequence of that flight's footprints covering a higher fraction of waterlogged environments.

The mean flux for all flights is 0.043 µg $N_2O$ m$^{-2}$ s$^{-1}$ (Table 2). This average is significantly different from zero flux ($p < 0.01$) as determined by both bootstrap-derived CIs and the student's *t*-test as described in the Methods section (see Table S1 for 99% CI ranges). This corresponds to a daily mean between 2.2-4.7 mg $N_2O$ m$^{-2}$ d$^{-1}$ (this range represents the 90% CI values in Table 2 being converted to mg $N_2O$ m$^{-2}$ d$^{-1}$). These observed $N_2O$ emissions are higher than expected (Zhuang et al. 2012). However, there have been several small-scale chamber studies that have observed $N_2O$ emissions within this mean daily range (Repo et al.
2009; Marushchak et al. 2011; Yang et al. 2018). Soil analyses on the North Slope thermokarst features in upland tundra have also found elevated soil $N_2O$ concentrations sustained throughout the growing season (Abbott et al. 2015). These elevated levels were attributed to abrupt thaw processes known as thermokarst, which can cause permafrost to collapse. This displaced soil redistributes soil organic matter into both oxic and anoxic environments, a condition conducive to producing $N_2O$ as a final product instead of as a metabolic intermediate for $N_2$ (Abbott et al. 2015). Transitions, in general, from oxic to anoxic
environments (and vice versa) are well-known to induce spikes in $N_2O$ emissions for a variety of soils (Schreiber 2012).

If results from the last week of August are representative of the whole month, the $N_2O$ emission over the span of August is ~0.04-0.09 g N m$^{-2}$. This range contains what is currently assumed to be the maximum emission over an entire year at these latitudes (~0.05 g N m$^{-2}$ yr$^{-1}$) (Anderson et al. 2010; Zhuang et al. 2012). Past static chamber studies and soil studies that
measured elevated permafrost $N_2O$ production observed that it was sustained throughout the entire growing period, which spans several months. One of these studies (Abbott et al. 2015) specifically examined permafrost in the North Slope, same as in our campaign. Furthermore, these studies largely attribute the elevated rate of soil $N_2O$ production to higher soil temperatures (Repo et al. 2009; Abbott et al. 2015). Soil temperatures taken near our flux tower were lower during our observation period than in previous weeks of August 2013 (Fig. S2).


**4 Discussion**

A body of evidence suggests the nutrient composition of, and microbial communities within, permafrost soils can be conducive
to nitrous oxide production. Boreal peat soils are known to have negligible $N_2O$ emissions when soil C/N ratios are above 25 (Klemedtsson et al. 2005). However, below this threshold, $N_2O$ fluxes can increase quite rapidly with decreasing C/N ratios. For the upper 3 m of permafrost, all three subsets of permafrost soils (histel, turbel, and orthel) have mean C/N ratios below 25, with turbel soils averaging the lowest at ~15 (Harden et al. 2012). Importantly, these values are averaged over many studies; the C/N ratio is highly variable. For example, eight yedoma (organic-rich permafrost soil) and thermokarst sites in Arctic Siberia were
reported to have C/N ratios of 11, averaged over 3 m depth (Fuchs et al. 2017). To reiterate, the upper 3 m is relevant for permafrost collapse, which routinely exposes deeper soil to the atmosphere. Furthermore, as discussed in a review of permafrost microbiology by Jannson et al. (2014), metagenomic analyses (analyses quantifying relative abundance of genes in an environmental sample) performed on permafrost cores suggest $N_2O$ is likely the final product for denitrification. This is because while most of the genes for the denitrification pathway were observed, the relative abundance of genes corresponding to the final
steps was determined to be too low to lead to any significant conversion of $N_2O$ to $N_2$ (Jannson et al. 2014).

It is unclear whether these observed emissions signify a recent trend or have been constant over time because the data collected on permafrost $N_2O$ fluxes are severely limited. No estimate of permafrost $N_2O$ emissions before the Industrial Revolution exists, and few data have been collected since (Davidson et al. 2014). Therefore, it is unknown how these emissions have changed since
global climate change started significantly affecting the permafrost landscape. However, it is well established that the troposphere at these higher latitudes has warmed, on average, 1.9 times more than the global average (Serreze et al. 2011). Soil temperatures have increased as well. Temperatures of permafrost soils in Northern Alaska, for example, increased by up to 3 °C since the 1980s (IPCC 2013). Permafrost warming/thaw, independent of permafrost collapse, has been demonstrated to increase $N_2O$ emissions significantly (Elberling et al. 2010; Voigt et al. 2016; Voigt et al. 2017). Furthermore, this temperature increase
has induced permafrost degradation over time, which has manifested as the expansion of thermokarst features shown to promote elevated $N_2O$ production even further as discussed above. Finally, increased permafrost thaw may make soil drainage more efficient, thus reducing the extent of waterlogged environments in higher latitudes (Avis et al. 2011). More streamlined draining also allows for a greater extent of draining and rewetting of permafrost soils, a process shown to increase thawed permafrost emissions of $N_2O$ ten-fold (Elberling et al. 2010). All of the observed changes to permafrost discussed above are variables that
have been examined with respect to nitrous oxide emissions, and they have all have been shown to increase the fluxes of this gas. Based on this existing literature, our observed $N_2O$ emissions may reflect a positive climate feedback already in progress. That being said, it is unclear to what extent future emission rates will increase because soil temperature is only one of many factors that will continue to change at high latitudes, with increasing vegetation being the most likely to negate the effects of increasing soil temperatures (Repo et al. 2009; Voigt et al. 2017).

## 5 Conclusion

In this campaign, we flew over ~310 $km^2$ of the Alaskan North Slope and measured $N_2O$ flux using the airborne Eddy Covariance technique. We observed spotty spatial distribution of elevated $N_2O$ emissions that averaged to 0.043 (0.025, 0.055) µg $N_2O$ $m^{-2}$ $s^{-1}$. These results corroborate several recent studies that used the static chamber method and observed permafrost soils emitting significant levels of nitrous oxide that are sustained throughout the entire growing period. This is a contrast to the traditional view regarding emissions at these latitudes. Importantly, we corroborate these findings in a complementary way: by observing fluxes on a landscape scale rather than the much smaller-scale soil plots in chamber studies, which are intended more to understand temporal representativeness and mechanisms of $N_2O$ production. While our study spans a spatial coverage greater by orders of magnitude than any previous study, it is still preliminary. The Arctic/sub-Arctic covers a vast area, and our observations do not necessarily represent the entire, months-long growing period. This limitation notwithstanding, we demonstrate that it is possible to apply the established airborne EC technique to the trace gas nitrous oxide to more thoroughly evaluate emissions of $N_2O$ in permafrost regions. This approach is a useful supplement since most of the landscape is remote and inhospitable, making maintenance of flux towers and chambers intractable.

Climate projection models and stratospheric ozone depletion assessments rely on global $N_2O$ budgets to predict future atmospheric scenarios (Ravishankara et al. 2009; Meinshausen et al. 2011). Considering the observed $N_2O$ fluxes reported here, more field campaigns that employ airborne EC technique or similar measurement techniques designed for much larger spatial coverage should be employed. These should also be coupled with ground-based measurements that can help corroborate airborne findings and better pinpoint where elevated $N_2O$ emissions might be occurring. Finally, future research efforts should consider following in the monitoring footsteps of $CO_2$ and $CH_4$ and establish continuously measuring EC towers for permafrost $N_2O$. Campaigns like these would help better determine whether the current, data-limited assumption of negligible $N_2O$ emissions is a correct one. If permafrost $N_2O$ emissions are already not negligible, their predicted increase with warming permafrost soil temperatures could result in a noncarbon climate feedback of a currently unanticipated magnitude.

**Data Availability.** The datasets generated and analyzed during the current study are available at: Sayres, D., Dobosy, R. Alaska 2013 Campaign. *Harvard Dataverse, V1*. doi: 10.7910/DVN/YM70Y7 (2018).

**Author contributions**

B.B., R.D., D.S.S, and J.G.A. designed the study. D.S.S., E.D., and C.H. contributed to instrument development, data collection and field work. J.W. and R.D. contributed to data processing and uncertainty analysis. J.W. was mainly responsible for interpreting the results and writing the manuscript. All authors participated in writing/editing the manuscript.

**Competing interests**

The authors declare that they have no conflict of interest.

**Acknowledgements**

We thank M. Rivero, N. Allen, and C. Tuozzolo for their laboratory and field assistance; Bernard Charlemagne for piloting the aircraft; J.B. Smith, E. Moisson, and A. Bendelsmith for their comments on the manuscript. This work was funded by NSF grant 1203583.

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

## Figures and Tables

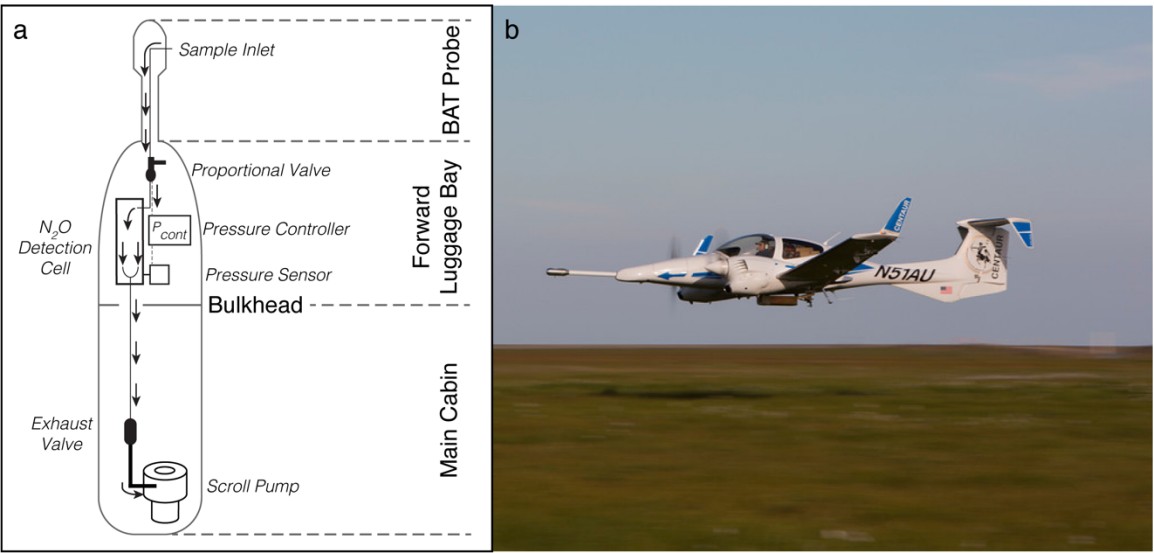

**Figure 1.** FOCAL during flight. **a**, Top-down schematic of the atmospheric gas flow through the aircraft (not to scale). The sample inlet is located on the BAT probe, located at the nose of the plane. The gas is pumped through the pressure-regulated detection cell of the ICOS spectrometer, located within the luggage bay in front of the pilot. **b**, Image of the Diamond DA-42 flying around 15 m above the surface.

25

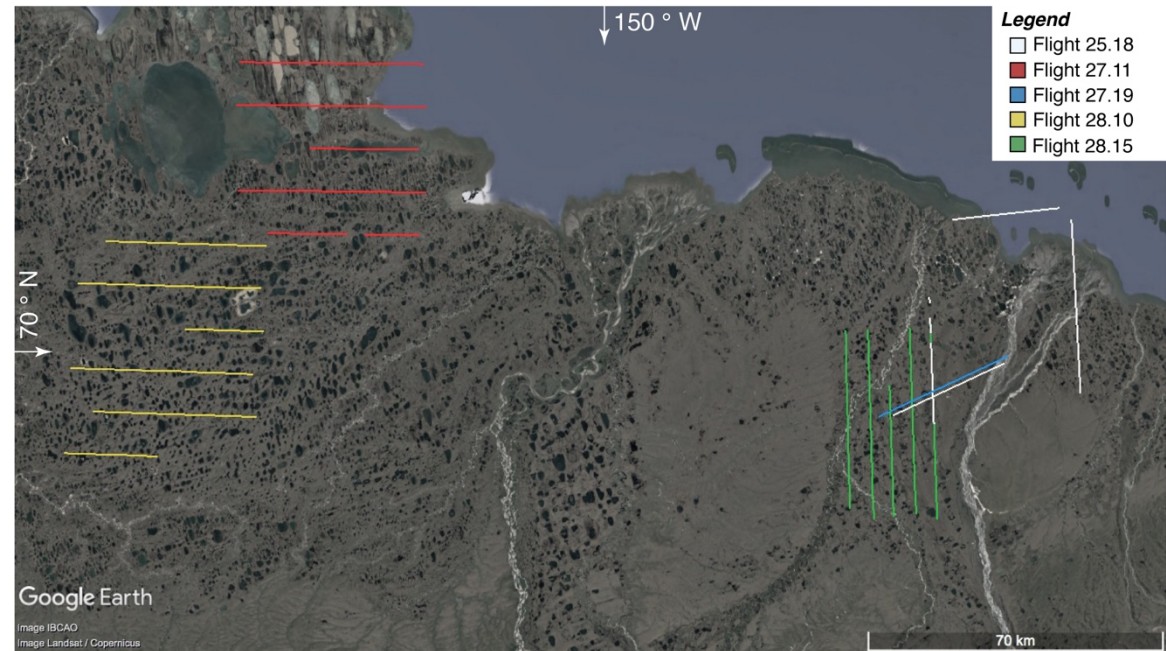

**Figure 2.** Flight tracks for August 2013 campaign where the paths represent sections of the flights that were suitable for flux calculations (aircraft flying on a straight, level path below 50 m). (Map image credit: Google Earth).

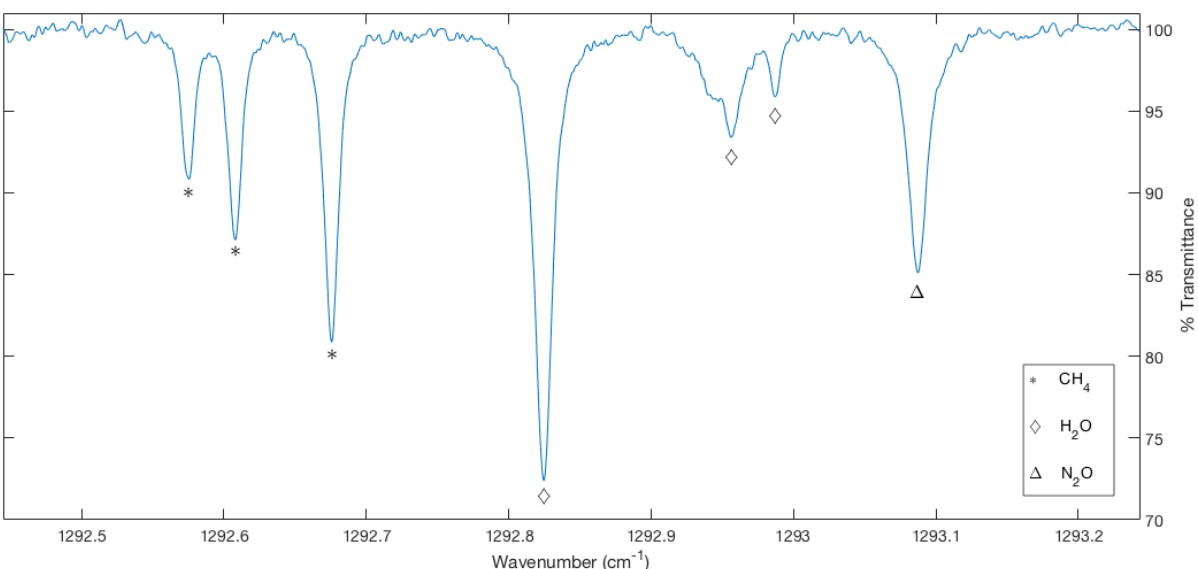

**Figure 3.** Sample 10-Hz ICOS spectrum taken from Flight 28.10.

25

**Table 1. Description of August 2013 Flights.** Flight date is the day of the flight in August (DD) and middle time of flight to nearest hour (HH). Temperature is the air temperature measured during the flight, averaged over all measurements made below 100 m. The dominant land classes are listed in order of decreasing relative contribution to the observed fluxes, determined by footprint analysis coupled with a landcover map. (FWM, Fresh Water Marsh).

| Flight date DD.HH | Start time UTC - 10 | End time UTC - 10 | Temperature (°C) | Dominant land classes |
|---|---|---|---|---|
| 25.18 | 17:43 | 19:49 | 5 | Sedge (33%), Mesic sedge (19%), FWM (8%), Open Water (7%) |
| 27.11 | 09:40 | 13:00 | 6 | Open Water (31%), Sedge (26%), FWM (17%), Tussock Tundra (14%) |
| 27.19 | 16:46 | 20:02 | 10 | Sedge (44%), Open Water (16%) Mesic sedge (15%), FWM (15%) |
| 28.10 | 08:39 | 11:39 | 11 | Tussock tundra (46%), Open Water (26%), Sedge (19%), FWM (7%) |
| 28.15 | 13:59 | 15:44 | 16 | Sedge (47%), Mesic sedge (26%), FWM (8%), Open Water (7%) |

25

**Table 2: Observed flux averages.** Area covered is the footprint scope of the measurements made for each flight. Spatially averaged fluxes are presented with bootstrap-derived 90% confidence intervals in parentheses. Asterisks indicate mean flux is significantly greater than 0 $\mu g$ $N_2O$ $m^{-2}$ $s^{-1}$ ($p < 0.01$).

| Flight date DD.HH | Area covered (km²) | Mean $N_2O$ flux ($\mu g$ $N_2O$ $m^{-2}$ $s^{-1}$) |
|---|---|---|
| 25.18 | 90 | 0.05* (0.031, 0.082) |
| 27.11 | 86 | -0.01 (-0.035, 0.028) |
| 27.19 | 22 | 0.015 (0.004, 0.032) |
| 28.10 | 69 | 0.10* (0.068, 0.140) |
| 28.15 | 44 | 0.04 (0.005, 0.080) |
| All flights | 311 | 0.043* (0.025, 0.055) |

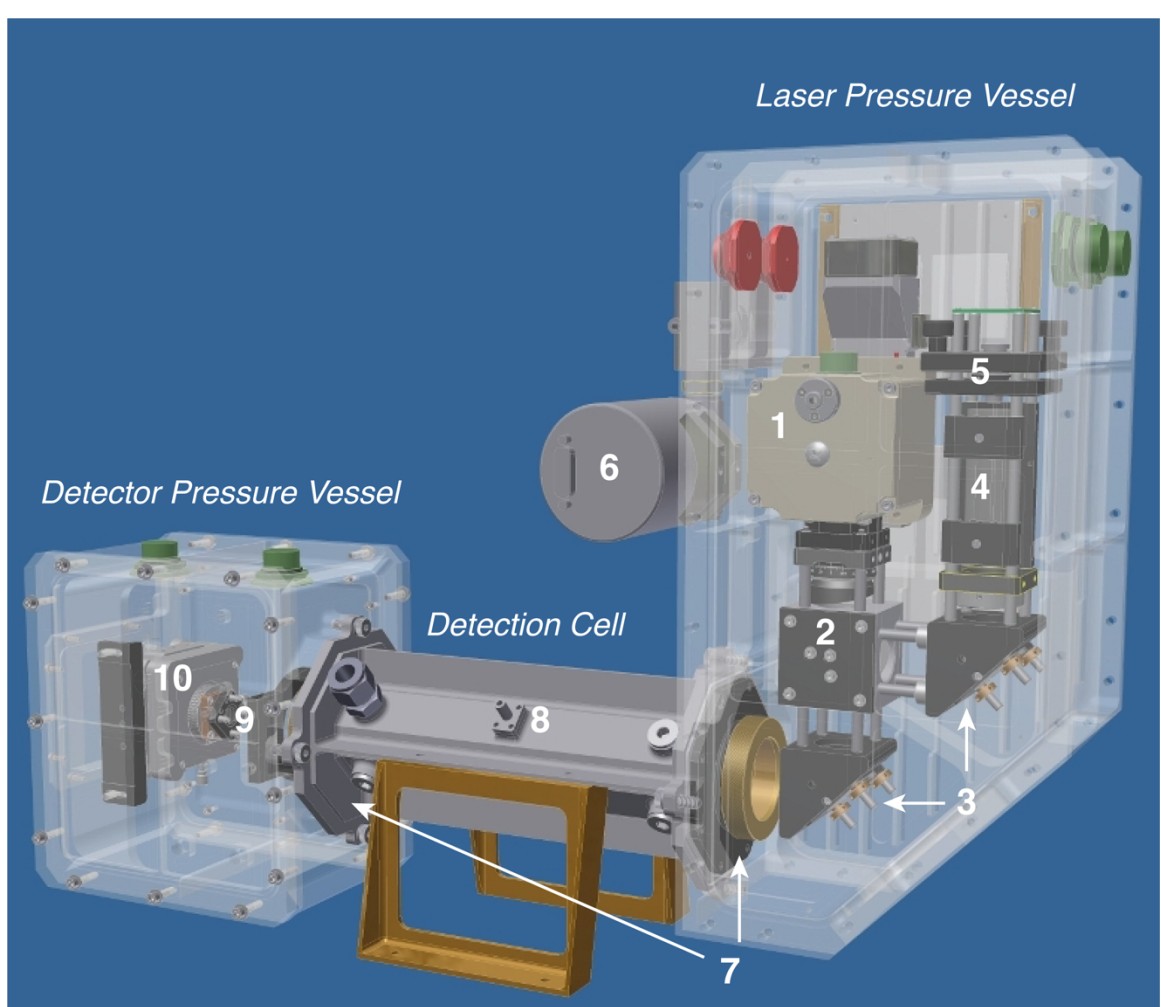

**Figure 4.** CAD model of N$_2$O/CH$_4$/H$_2$O ICOS instrument shows 1) quantum cascade laser housing, 2) beam splitter, 3) steering optics, 4) Ge etalon, 5), etalon detector, 6) Baratron pressure sensor, 7) ICOS cavity mirrors, 8) temperature sensor port, 9) focusing optics, and 10) MCT detector. The detection cell is 25 cm long.

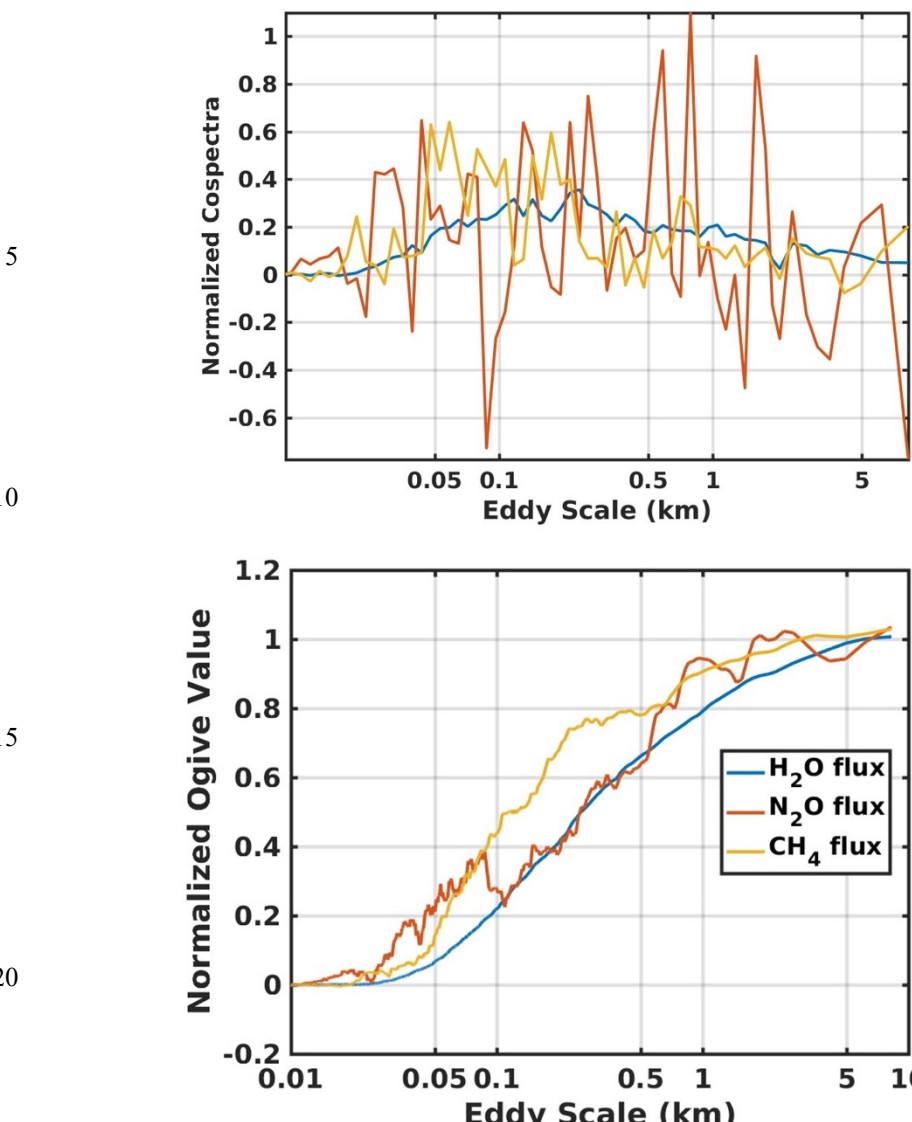

**Figure 5.** Normalized average cospectra and ogives for $H_2O$ flux, $N_2O$ flux, and $CH_4$ flux. The average covers the entire flight campaign. The ogive integration starts from the small eddy sizes.

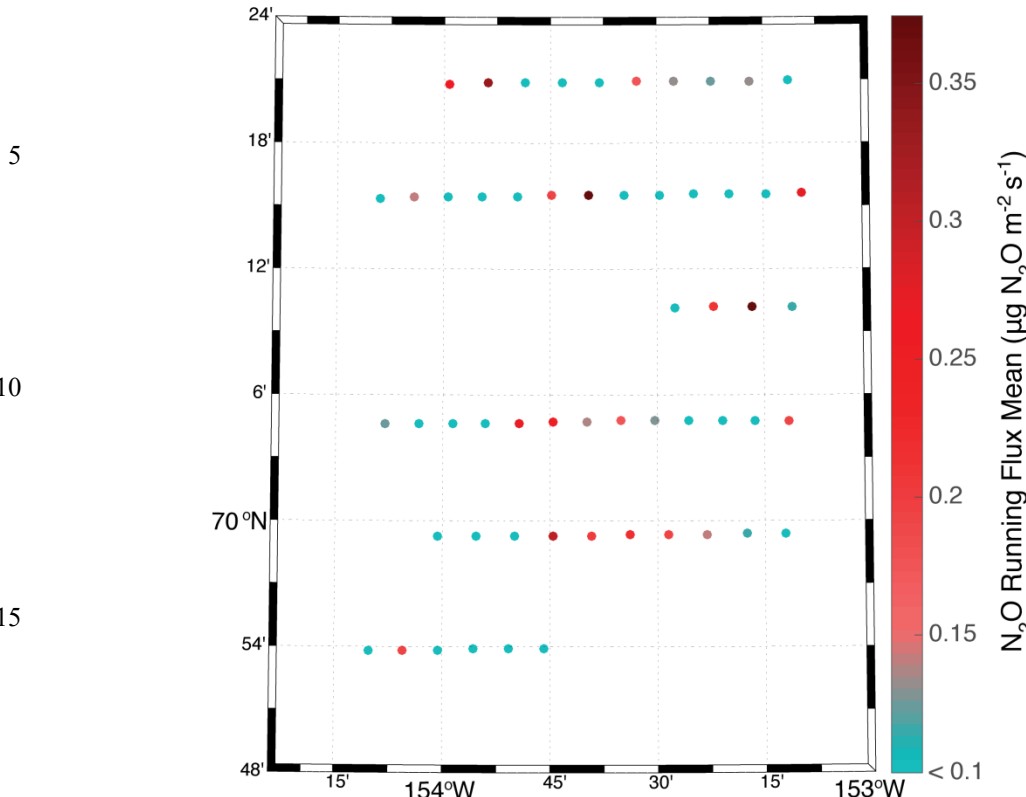

**Figure 6.** Spatial flux map for Flight 28.10, where the circles on the map each represent the $N_2O$ flux averaged over 6 km with 3 km overlap. Values within $\pm$ 0.1 $\mu$g $N_2O$ m$^{-2}$ s$^{-1}$ are treated as zero.