# Peer review of "Figure S1. Image of N2O/CH4/H2O instrument in aircraft's luggage bay, which shows 1) electronics board, 2) detector pressure vessel, 3) detection cell, and 4) laser pressure vessel. Electronic cables were removed for clarity."

_Atmospheric Chemistry and Physics, 2018_

## Referee Comment (RC1) · Anonymous Referee #1 · 11 Nov 2018

General comments

The results from studies conducted during last ten years suggest that not only fluxes of carbon dioxide and methane but also nitrous oxide fluxes have to be considered when evaluating the present and future atmospheric impact of permafrost regions. It is well known that nitrous oxide fluxes have generally high spatial variation. The studies by chamber methods have shown that the nitrous oxide emissions from permafrost regions have extremely high spatial variation. Therefore, all different functional surfaces should be included to the measurements done by chambers and high resolution tools are needed for mapping of various surfaces to upscale the nitrous oxide emissions

from permafrost landscape. Eddy covariance (EC) method integrates gas fluxes for a larger area and offers an alternative tool to determine gas fluxes for landscape/region. However, to set tower based EC instrumentation in remote regions, like permafrost regions, is a demanding task. In the present study airborne eddy covariance method was used first time to measure nitrous oxide fluxes in permafrost region. The study is an important contribution to the ongoing efforts to evaluate the importance of nitrous oxide fluxes in the permafrost regions. The authors have considered in details the methodological aspects of the airborne EC method they applied. The mean nitrous oxide emission for the 310 km2 area they report is high when compared to the emission rates obtained by chamber techniques from permafrost soils. Such high emission rates are measured from high-emitting patches in permafrost regions. The footprint analyses here also indicated areas with negligible emissions and areas with high nitrous oxide emissions.

As the authors noted there is a study (Abbott et al. 2015. Global Change Biology 21: 4570-4587) showing that permafrost collapse in the study region, North Slope of Alaska, increases nitrous oxide content in soil. To get such a high mean emission rate shown here by the EC, the emissions from the high-emitting areas have to be very high. Would be excellent if the authors could get some published or non-published data on nitrous oxide emissions in the region based on chamber measurements or determined by a gas gradient approach based on nitrous oxide content in soil. This data could then be upscaled by estimating the total coverage of the high emitting areas. If the nitrous emissions from these analyses are in the same range as the mean emission rate here, this could confirm the results obtained by EC.

Some detailed comments

Page 2/line 1 Change the text to "…However, recent in situ measurement of permafrost soils in Russian tundra and northern Finland (Repo et al. 2009; Marushchak et al. 2011)"

Page 2/lines 10-16 The discussion on the flux data generated by chamber method could be modified to state that there are both disadvantages and benefits using chamber method for the gas fluxes. By the chambers e can catch efficiently the various functional surfaces, even very small. So, we can get knowhow on the soil and vegetation related factors affecting gas fluxes. To obtain landscape or regional fluxes by chambers for permafrost regions, accurate distribution of the functional surfaces is required. This can be done using e.g. satellite images (e.g. Treat et al. 2018. Global Change Biology, Doi: 10.1111/gcb14421).

---

## Referee Comment (RC2) · Anonymous Referee #2 · 15 Nov 2018

This manuscript presents an airborne Eddy Covariance technique to measure N2O emissions from permafrost regions. A summer flight campaign was performed in August 2013 above North Slope, Alaska. The measurements from this airborne laboratory gave an average N2O flux of 3.8 mg N2O m-2 d-1, higher than the expected range for permafrost regions. The authors speculate that increasing thermokarst under climate change (a hotspot of N2O emission in permafrost) could have resulted in such high N2O emissions. Overall, the analytical technique and estimation of N2O flux based on EC Method are well described with plenty of details. The analytical precision of N2O mixing ratios seems satisfying for airborne measurement, and uncertainty in flux estimations has been also discussed. However, given the unexpectedly high N2O flux,

it remains a question to us how realistic flux data we could obtain based on these airborne measurements within limited time scales. Although this approach has been tested for CH4/H2O with a near-by EC tower (Sayres et al., 2017), spatial variabilities in N2O fluxes could largely differ. Therefore, more ground-based fluxes data in the near region by EC tower or chamber measurements are necessary to confirm the applicability of this technique. Alternatively, if N2O emission factors or edaphic parameters are available in this or other similar regions, a ground-based model estimation of N2O fluxes separating landscape elements may strengthen the whole manuscript.

Detailed comments:

Abstract: Some explanation of the high N2O fluxes needs to be implemented. Also, please indicate the site location in the abstract.

Page 2, Line 10-16: The authors argued that chamber measurement or lab studies cover small spatial scales. However, the airborne measurements cover only short time periods. Perhaps a little more background on spatio-temporal variabilities in N2O fluxes from permafrost?

Page 2, Line: 17-20: Much of the detailed information on flight campaign could be put in M&M.

Page 5, Line 1, equation (3): I think that running flux method (RFM, Sayres et al. 2017) was used for N2O flux estimations in this manuscript. However, Sayres et al. (2017) suggested advantage of flux fragment method (FFM) against RFM in their airborne EC CH4 study. Also, they claimed that FFM can isolate flux contributions from individual surface land classes. Please explain it.

Page 7, Line 29 and Line 32: Could you give a more quantitative description of land classes (in % or area size) for Table 1?

Page 8: Line 27-30: Gene abundance does not directly refer to denitrification and N2O reduction rates. It still needs to be expressed so that N2O can be reduced. Better

focus on the O2 inhibition effect on N2O reductase.

---

## Author Comment (AC1) · 24 Jan 2019

Response
1. Referees 1 & 2 Comments

Referee 1*: The study is an important contribution to the ongoing efforts to evaluate the importance of nitrous oxide fluxes in the permafrost regions. The authors have considered in details the methodological aspects of the airborne EC method they applied…The footprint analyses here also indicated areas with negligible emissions and areas with high nitrous oxide emissions.*

Referee 2*: Overall, the analytical technique and estimation of N2O flux based on EC Method are well described with plenty of details. The analytical precision of N2O mixing ratios seems satisfying for airborne measurement, and uncertainty in flux estimations has been also discussed.*

1.1 Response to Referees: We appreciate the referees' affirmation of the techniques and uncertainty analysis on which we base our conclusions.

1.2 Change to Manuscript: None.

2. Referees 1 & 2 Comments

Referee 1: *To get such a high mean emission rate shown here by the EC, the emissions from the high-emitting areas have to be very high. Would be excellent if the authors could get some published or non-published data on nitrous oxide emissions in the region based on chamber measurements or determined by a gas gradient approach based on nitrous oxide content in soil. This data could then be upscaled by estimating the total coverage of the high emitting areas. If the nitrous emissions from these analyses are in the same range as the mean emission rate here, this could confirm the results obtained by EC.*

Referee 2: *However, given the unexpectedly high N2O flux, it remains a question to us how realistic flux data we could obtain based on these airborne measurements within limited time scales. Although this approach has been tested for CH4/H2O with a near-by EC tower (Sayres et al., 2017), spatial variabilities in N2O fluxes could largely differ. Therefore, more ground-based fluxes data in the near region by EC tower or chamber measurements are necessary to confirm the applicability of this technique. Alternatively, if N2O emission factors or edaphic parameters are available in this or other similar regions, a ground-based model estimation of N2O fluxes separating landscape elements may strengthen the whole manuscript.*

2.1 Response to Referees: Indeed the high-emitting "hot-spots" have to be quite prominent to affect the overall average. The nature of an airborne study is to provide a spatial survey of the prevalence and spatial distribution of such high-emission locations along with any other distributed sources in an area difficult of access. The reviewer characterizes the high spatial variation of $N_2O$ flux to be well known from chamber methods. The significance of our result, in addition to confirming the spottiness, is to find such hot spots and other sources of $N_2O$ to be sufficiently strong and/or numerous over landscape and larger scales on the North slope to add up to average $N_2O$ emission comparable to that found in the tropics.

To our knowledge, established emission factors do not currently exist for permafrost $N_2O$ emissions. No significant emissions were reported from any permafrost land class until 2009, when researchers identified bare peat circles as potential emitters of

significant amounts of $N_2O$. Several other studies (cited in the manuscript) have since come out reinforcing the notion that $N_2O$ could be significant. However, these are mostly laboratory studies, soil $N_2O$ observations, and metagenomic analyses, which do not translate to emission factors. The first results from this airborne approach reinforce the need for continued and enhanced examination, both spatially and temporally extensive, in the arctic. We seek to publish the significant result we've observed on a landscape-scale to help motivate future studies that can provide the confirmation sought in Referee 1's comment.

In response to Referee 2, we agree that the spatial variability for N2O could be different -- as demonstrated by our measurements. However, that does not necessitate comparative ground-based measurements catered specifically to N2O. Comparison with another measurement technique would only be necessary if there are concerns about our primary method. No concerns were given. As stated in the manuscript, our comparison with a nearby tower demonstrates that our instrument is capable of measuring airborne fluxes of gases. We also calibrated our EC system both in a wind tunnel and during the campaign. From the perspective of making trace gas flux measurements then, the only difference between $CH_4$, $H_2O$, and $N_2O$ is a different absorption feature on our spectra. Since we demonstrate that the template of our EC system works, all that is left is making sure the $N_2O$ sensor works properly. We have done this, both in lab and in flight, and furthermore, performed an uncertainty analysis on the N2O flux data, which includes a comprehensive ogive plot. The referee seems satisfied by this as shown in Comment 1 above.

Separately, there is an issue regarding Referee 2's recommendation that we use chambers to confirm airborne measurements. For an airborne measurement to correspond to a fixed surface measurement their footprints must correspond. A chamber measures the same small patch (typical size for a chamber is ~1 $m^2$). By contrast, the minimum spatial coverage of an aircraft measurement, determined by the largest scale of atmospheric turbulence, is closer to 6 $km^2$. For the footprints from the two methods to correspond, the aircraft track must lie over a homogeneous surface of area 6 $km^2$ having the same character as that being sampled by the chamber (and the chamber's 1-square-meter footprint must accurately represent the 6-square-kilometer surface covered by the aircraft's measurements). For $CH_4$ or $CO_2$ such a surface can be identified by remote sensing. As discussed above, however, no comparable surface classification for $N_2O$ is available to our or, presumably, to the referees' knowledge. An EC tower would sample a larger surface area than a chamber but would be subject to the same issues in establishing a common footprint with aircraft for $N_2O$.

Therefore, while we do agree that comparison with a tower for N2O specifically 'would be excellent' as Referee 1 states, we disagree that it's necessary as Referee 2 suggests. Having said that, we do state on Page 9/lines 20-21 that this is a stepping stone where future research would provide that ground comparison. Considering no one is currently doing this, we strongly believe publishing this information for the scientific community to see is an important step to motivate the community to look further into what's going on here. This is especially true since several chamber-based

publications have already recently been published suggesting the community's assumptions about permafrost N2O emissions may need to be refined.

Finally, we agree with Referee 2 that we cover a smaller time scale than a longer-term ground-based measurement. But accordingly, we restrain our extrapolation to the month of August instead of the entire summer precisely because of our shorter time scale.

2.2 Change to Manuscript: None.

3. Referee 1 Comment
*Page 2/line 1 Change the text to ". . .However, recent in situ measurement of permafrost soils in Russian tundra and northern Finland (Repo et al. 2009; Marushchak et al. 2011)"*
3.1 Response to Referee: We agree with the suggestion and have changed the manuscript accordingly.
3.2 Change to Manuscript: The text was changed as Referee 1 suggested (Page 2/line 3).

4. Referee 1 Comment
*Page 2/lines 10-16 The discussion on the flux data generated by chamber method could be modified to state that there are both disadvantages and benefits using chamber method for the gas fluxes. By the chambers e can catch efficiently the various functional surfaces, even very small. So, we can get knowhow on the soil and vegetation related factors affecting gas fluxes. To obtain landscape or regional fluxes by chambers for permafrost regions, accurate distribution of the functional surfaces is required. This can be done using e.g. satellite images (e.g. Treat et al. 2018. Global Change Biology, Doi: 10.1111/gcb14421).*
4.1 Response to Referee: Thank you for pointing this out. We did not bring up the spatial limitation of the chamber method to suggest that method has no benefits. Rather, our goal was to point out a gap in the research that an alternative method could help fill in – our method specifically. We did point to several benefits of chamber studies on Page 2/line 10. Still, we appreciate that a reader could get the impression we might be offering airborne EC as a replacement method. That is not what we are doing. Airborne EC complements the chamber method and ground-based measurements in general. We will alter the text to better convey this sentiment.

As for the latter part of the comment, it's true that research teams perform quite spatially extensive extrapolations from chamber measurements to get a landscape-scale estimate of some trace gas emissions. Treat et al. 2018, the paper the referee cites, examines this method, and the potential pitfalls of using it, for $CO_2$ and $CH_4$. It does not for $N_2O$. This is probably because permafrost $N_2O$ emissions have had significantly less research effort dedicated to them. Unlike the EC tower network that exists to varying extent for $CO_2$ and $CH_4$, there is not a single EC tower that provides consistent, long-term measurements of permafrost $N_2O$ emissions. Chamber studies are also quite sparse. While research effort sufficient for discussion of this type of extrapolation may eventually be put forth for $N_2O$, such discussion feels more relevant for $CO_2$ and $CH_4$ at this time.

We mentioned that Treat et al. discuss potential pitfalls of this type of extrapolation in their paper. We'd like to touch on that again as a way to further emphasize how it might be premature to discuss these types of extrapolations for permafrost $N_2O$. One of Treat's main messages is that satellite images can cause severe underestimations if an insufficient resolution is chosen. They've found that you need a better spatial resolution for $CH_4$ than for $CO_2$ because $CH_4$ emissions are more spatially variable. Without proper resolution, the $CH_4$ emissions are severely underestimated (by up to 65%). $N_2O$ is considered much more variable and spotty than $CH_4$. We can imagine the spatial resolution therefore needs to be even better to extrapolate $N_2O$ emissions, but it's unknown what this resolution should be or whether traditional satellite maps could properly accomplish this (the land cover map we used has the recommended 30 m x 30 m resolution for $CH_4$). Our presented results involved no extrapolation and, therefore, no need to make assumptions about the characteristics of a particular land class as it relates to $N_2O$ or to worry about potential issues with the resolution of satellite images used to make the extrapolations. Instead, we obtained a landscape-scale estimate from hours of airborne measurements spanning hundreds of square kilometers of the Arctic. There are certainly advantages of the chamber method over airborne EC, as we have now more explicitly stated. However, obtaining a landscape-scale estimate is not one of them.

4.2 Change to Manuscript: Page 2/lines 14-23 changed to "The past studies on permafrost $N_2O$ emissions have provided insight into the mechanisms of the gas's production and subsequent release into the atmosphere. The studies have been either laboratory studies or ground-based chamber studies. In general, chamber studies have the advantage of observing the same site for relatively long time periods. Additional variables (e.g. pH, water saturation) can be monitored, too, which are crucial to understanding how that environment might influence the observed extent of $N_2O$ emissions. However, each chamber covers around 1 $m^2$, and a feasible chamber study can only entail a limited number of sites. Consequently, past observations have covered extremely small areas – less than 50 $m^2$ (Repo et al. 2009; Marushchak et al. 2011; Yang et al. 2018). Therefore, the landscape scale of this phenomenon remains unknown, let alone the regional and continental scales. Landscapes deemed vulnerable to thaw-induced $N_2O$ emissions, permafrost and thermokarst regions, cover about one fourth of the Arctic/sub-Arctic (Voigt et al. 2017). One of those vulnerable areas is the Alaskan North Slope, which is the focus of this study. To get a landscape-scale estimate of the magnitude of permafrost $N_2O$ emissions during late summer, we measured $N_2O$ fluxes over the North Slope in late August 2013 using the airborne eddy covariance (EC) technique."

We also modified the text in the Conclusion (Page 9/lines 14-16), to read: "Importantly, we corroborate these findings in a complementary way: by observing fluxes on a landscape scale rather than the much smaller-scale soil plots in chamber studies, which are intended more to understand temporal representativeness and mechanisms of $N_2O$ production."

5. Referee 2 Comment

   *Page 2, Line 10-16: The authors argued that chamber measurement or lab studies cover small spatial scales. However, the airborne measurements cover only short time periods. Perhaps a little more background on spatio-temporal variabilities in N2O fluxes from permafrost?*

   5.1 Response to Referee: The reported average presented in our manuscript represents, in total, a little under 10 hours of airborne measurements across the North Slope semi-randomly sampled over the span of a week. To give an estimate for the entire month of August, we therefore extrapolate our data by an order of 100. By comparison, a typical chamber study covers an area around 10,000,000 times smaller than our spatial coverage. We agree that we are temporally limited, but the spatial limitation for chambers seems more severe. Regardless, our point was not to suggest airborne EC is better than chamber studies, merely that it's more appropriate for establishing a landscape-scale estimate. We have modified the text to better convey this.

   5.2 Change to Manuscript: Please see changes in Comment Response 4

6. Referee 2 Comment

   *Page 2, Line: 17-20: Much of the detailed information on flight campaign could be put in M&M.*

   6.1 Response to Referee: Most of that text has been moved to the first paragraph of the Methods section.

   6.2 Change to Manuscript: Except for the last sentence, Page 2, Lines 17-20 were all moved to the first paragraph of Methods section (Page 2, Line 35-37). The text was also slightly modified to better flow with the text following its new location.

7. Referee 2 Comment

   *Page 5, Line 1, equation (3): I think that running flux method (RFM, Sayres et al. 2017) was used for N2O flux estimations in this manuscript. However, Sayres et al. (2017) suggested advantage of flux fragment method (FFM) against RFM in their airborne EC CH4 study. Also, they claimed that FFM can isolate flux contributions from individual surface land classes. Please explain it.*

   7.1 Response to Referee: The running flux method was not used for $N_2O$ flux estimations presented in Table 2 of this manuscript. Equation (3) is the general equation for airborne EC fluxes, not an equation specific to RFM. We don't use FFM either. We opted instead for the more robust approach of averaging over entire flights in order to focus on the overarching landscape and to present simpler, more statistically sound results.

   We do, however, use RFM to determine the values for the data in Figure 6. RFM is an application of equation (3), as described in Page 7/lines 9-10 of the ACPD manuscript (note we avoid using the term 'RFM' in an effort to minimize jargon in the manuscript). Regarding FFM versus RFM, Sayres et al. 2017 only suggests advantage of FFM over RFM under certain circumstances. They do not assert FFM has an overall

advantage (some of the authors of that publication also author this manuscript). The goal of Figure 6 is to illustrate the spottiness of $N_2O$ emissions, and RFM is better suited for that purpose.

7.2 Change to Manuscript: On Page 5/line 2, we changed the equation (3) description from 'standard equation' to 'general equation' to better clarify this is not an equation specific to RFM or FFM.

8. Referee 2 Comment
*Page 7, Line 29 and Line 32: Could you give a more quantitative description of land classes (in % or area size) for Table 1?*

8.1 Response to Referee: We have provided percent of observed footprints that fall into those land classes. In order to quantify based on land type, we more strictly followed the NSSI Landclass map; lakes and rivers have been combined to open water as they are according to NSSI.

8.2 Change to Manuscript: See imbedded table below.

| Flight date DD.HH | Start time UTC - 10 | End time UTC - 10 | Temperature (°C) | Dominant land classes |
|---|---|---|---|---|
| 25.18 | 17:43 | 19:49 | 5 | Sedge (33%), Mesic sedge (19%), FWM (8%), Open Water (7%) |
| 27.11 | 09:40 | 13:00 | 6 | Open Water (31%), Sedge (26%), FWM (17%), Tussock Tundra (14%) |
| 27.19 | 16:46 | 20:02 | 10 | Sedge (44%), Open Water (16%) Mesic sedge (15%), FWM (15%) |
| 28.10 | 08:39 | 11:39 | 11 | Tussock tundra (46%), Open Water (26%), Sedge (19%), FWM (7%) |
| 28.15 | 13:59 | 15:44 | 16 | Sedge (47%), Mesic sedge (26%), FWM (8%), Open Water (7%) |

9. Referee 2 Comment
*Page 8: Line 27-30: Gene abundance does not directly refer to denitrification and N2O reduction rates. It still needs to be expressed so that N2O can be reduced. Better focus on the O2 inhibition effect on N2O reductase.*

9.1 Response to Referee: Recognizing that gene abundance alone does not account for the fraction of the genes that are expressed in a population, the connection is at some level indicative of the population of the microbial community present. This argument is presented as one found in the literature that supports the possibility of N2O production instead of N2 production. Here's the quote from the microbiology review paper we cite for that assertion:

"In the Arctic permafrost metagenomes that have been analysed so far, most of the genes that are involved in the denitrification pathway have been detected, but the relative gene abundances for the last steps in the pathway were too low to lead to N2 production."

The review then goes on to posit that a possible consequence of this is the accumulation of N2O. Because this is not our argument, but an argument from a published review paper in the field of microbiology, we will keep the sentence as is.

9.2 Change to Manuscript: None.

10. Referee 2 Comment

*Abstract: Some explanation of the high N2O fluxes needs to be implemented. Also, please indicate the site location in the abstract.*

10.1 Response to Referee: We agree the site location should be in the abstract. We will add that along with the date. However, we feel that an explanation of the mechanism of the high N2O fluxes would be inappropriate since our results do not provide that insight, and explanations are largely based on insights provided by past chamber and laboratory studies. Having said that, we do provide more detail to our findings to clarify that our average represents observations with high variability.

10.2 Change to Manuscript: Page 1, Lines 15-17 has been altered as follows: "In late August 2013, we used the airborne eddy covariance technique to make *in situ* $N_2O$ flux measurements over the North Slope of Alaska from a low-flying aircraft spanning a much larger area: around 310 $km^2$. We observed large variability of $N_2O$ fluxes with many areas exhibiting negligible emissions."

---

## Author Response (AR2)

**Minor Revision Response**

Editor's Comment: In your final response, there are a few points brought up by one or even both of the reviewers where you had a lengthy discussion in your response but no change to the manuscript. Although in these cases it was generally down to different viewpoints, I would be glad if you can change your manuscript to reflect all the points you brought up in your author response and thus clarify your point of view.

Author response: This is fair. There are two comments where we responded but did not change the manuscript, Comments 2 and 9. For Comment 2, we have modified the introduction to more clearly convey the motivation for this research effort (Page 1/lines 37-40, Page 2/lines 14-20) and what particular benefits airborne EC brings to the study of permafrost N2O emissions (Page 2/lines 21-25). We added a bit more information to better explain the history of the airborne EC method as it applies to N2O (Page 2/lines 29-30, 34-37). We also slightly modified the conclusion to more explicitly acknowledge the time-scale limitation of our results (Page 8/lines 9-10) and offer up another possible avenue for future research efforts in monitoring N2O emissions that would be more temporally representative than an airborne campaign (Page 8/lines 19-20). For Comment 9, we modified the text to better clarify the assertion is not ours but one expressed in a separate, peer-reviewed publication (Page 7/lines 36-38).

**Permafrost Nitrous Oxide Emissions Observed on a Landscape Scale Using Airborne Eddy Covariance Method**

Jordan Wilkerson1, Ronald Dobosy2,3, David S. Sayres4, Claire Healy5, Edward Dumas2,3, Bruce Baker2, and James G. Anderson1,4,5

[revised manuscript text omitted]

**(Deleted: eddy covariance (EC)**

| Deleted: 1                                                | has been applied                              |  |  |
|-----------------------------------------------------------|-----------------------------------------------|--|--|
| Deleted: the trace gas Deleted: many times in the form of |                                               |  |  |
|                                                           |                                               |  |  |
| Deleted:                                                  | permafrost                                    |  |  |
|                                                           |                                               |  |  |
| Deleted: t                                                | to measure fluxes of the particular trace gas |  |  |
| Deleted:                                                  | gas fluxes                                    |  |  |

| Dele | eted: (Note that while the system measured N2O flux, the |
|------|----------------------------------------------------------|
| Dele | eted: also                                               |
| Dele | eted: ). N2O measurements                                |

The flights near the flux tower were performed to compare the airborne CH4 and H2O flux measurements with those from the EC flux tower (Dobosy et al. 2017). The CH4 and H2O fluxes agreed with the ground measurement, and the CH4 fluxes are consistent with other observed summertime permafrost CH4 emissions reported in the scientific literature (see Sayres et al. 2017). The only difference in the airborne flux measurements between CH4, H2O, and N2O is the particular absorption feature used 5 within the observed spectral region of the IR instrument, as further discussed in Section 2.2 (Fig. 3).

**2.1 BAT Probe Description and Calibration**

[revised manuscript text omitted]

| 5  | Measurement of H 2 O was calibrated using a dry-air tank coupled with a bubbler flow system as described in Weinstock et al. 2009. The H 2 O measurements were used to account for dilution and water-broadening effects on the N 2 O absorption feature and to convert the mixing ratio from moles per mole of total air to moles per mole of dry air for flux computation (Webb et al. 1980, Gu et al. 2012). The broadening coefficients were determined using the approach described in Rella 2010. Periodic in-flight calibrations were performed to track and correct for drift over the course of the flight (2 calibration cycles per flight). These were performed using a secondary standard (277 ppbv N 2 O) calibrated in lab to a WMO standard (Sayres et al. 2017). Before and after the campaign, calibrations were also conducted in lab using two primary WMO standards and a synthetic air tank (containing no N 2 O) to calibrate the absorption coefficient and check for linearity. The short-term precision of the ICOS instrument for N 2 O mixing ratios is determined using | Deleted: calibrations were performed |
|----|------------------------------------------------------------------------------------------------------------------------------------------------------------------------------------------------------------------------------------------------------------------------------------------------------------------------------------------------------------------------------------------------------------------------------------------------------------------------------------------------------------------------------------------------------------------------------------------------------------------------------------------------------------------------------------------------------------------------------------------------------------------------------------------------------------------------------------------------------------------------------------------------------------------------------------------------------------------------------------------------------------------------------------------------------------------------------------------------------------------------------------------------------------------------|--------------------------------------|
|    | $\sigma = \sigma_{1s} f_s^{-1/2} \tag{1}$                                                                                                                                                                                                                                                                                                                                                                                                                                                                                                                                                                                                                                                                                                                                                                                                                                                                                                                                                                                                                                                                                                                              | Deleted:                             |

[revised manuscript text omitted]